# The Influence of Remelting on the Properties of AlSi9Cu3 Alloy with Higher Iron Content

**DOI:** 10.3390/ma13030575

**Published:** 2020-01-25

**Authors:** Justyna Kasińska, Dana Bolibruchová, Marek Matejka

**Affiliations:** 1Department of Metal Science and Materials Technology, Kielce University of Technology, Al. Tysiąclecia Państwa Polskiego 7, 25 314 Kielce, Poland; 2Faculty of Mechanical Engineering, Department of Technological Engineering, University of Žilina, Univerzitná 8215/1, 010 26 Žilina, Slovakia; danka.bolibruchova@fstroj.uniza.sk (D.B.); marek.matejka@fstroj.uniza.sk (M.M.)

**Keywords:** Al-Si-Cu secondary aluminium alloy, remelting, iron intermetallic, metallography, deep etching, natural and artificial ageing, mechanical properties

## Abstract

This article aims to evaluate the influence of remelting on the experimental Al-Si-Cu type alloy with higher iron content on mechanical properties in relation to the resulting structure. The remelting or recycling process is one of the means of reducing the production costs of the forge plant. The experimental part deals with the analysis of the results of mechanical properties, structural analysis, and the process of crystallization of structural components and their changes due to the increased iron content caused by remelting at different states of the examined alloy. The effect of remelting and ageing on microstructure was observed using a combination of different analytical techniques (light microscopy, scanning electron microscopy (SEM), and upon deep etching and energy dispersive X-ray analysis (EDX)). Tensile strength and elongation tests point to the negative effect of alloying, a gradual increase in wt% Fe and a change in the morphology of the iron phases, which began to manifest significantly after the fourth remelting. The process of natural ageing has been shown to be effective only on alloys with a lower number of remelting cycles, whereas the application of artificial ageing has resulted in improved mechanical properties in all the test alloys.

## 1. Introduction

Recycling through multiple remelting is a major aspect in the continued lifetime of aluminum alloys. Metallurgy of recycled secondary aluminum alloys, as well as that of many other recycling processes, enables raw material savings and, in particular, energy savings with a profit of up to 95%. Another positive aspect of recycling aluminum alloys is the environmental impact, as only 5% of greenhouse gases are released in the production process compared to primary aluminum production [1,2,3]. Depending on the foundry, a few dozen percent of recycled material, such as gating systems, chips, or miscellaneous castings, is used today in the charge [4,5].

The recycled material in the batch will be increasingly used due to the economic and environmental benefits of these alloys in the casting process. A high amount of recycled aluminum material is also used for structural castings with complex shapes, e.g., automotive industry. It is impossible to trace the number of times that the specific portion of the material returns to the casting process. Despite the fact that this material meets all the required criteria given in EN 1706, there is “unwarranted” decrease in the mechanical properties after some time. It is an actual metallurgical topic that distresses foundries. Worldwide, there is a minimum of authors investigating this phenomenon in detail. There are authors [6] who state that aluminum can be recycled indefinitely without losing the desired properties associated only with a reduction in the content of some elements and a small change in microstructure and mechanical properties, which appears to be incorrect. However, there are several authors [7] who claim that with the increasing number of remelting, the alloy substantially loses its original properties and there is a significant degradation of structural components with a decrease in mechanical properties. The degradation of the structural components is in many cases unclear. Despite the knowledge gained so far on the effect of multiple remelting regarding the microstructure and the resulting mechanical properties, there is no clear opinion about its impact. For this reason, we have been studying the effects of remelting various Al-Si-Cu-based aluminum alloys for several years.

Due to the increasing use of recycled aluminum alloys for demanding castings, especially for the automotive industry, their quality is considered to be the key factor. The microstructure and mechanical properties of the alloy depend on a number of factors including its chemical composition. The most damaging element is generally considered to be iron, which, with other elements present in the alloy, can form intermetallic phases of a different morphology and length. Depending on the chemical composition of the alloy, the harmful effect of iron occurs when the so-called critical value of Fe_crit_ over-ranges in wt%, which is calculated using Equations [8,9,10,11]:Fe_crit_ ≈ 0.075 × (% Si) − 0.05,(1)

Several authors have reported the harmful effects of iron phases. In their work, the authors report that the presence of iron-based phases in the alloy structure results in a decrease in ductility and tensile strength. The cause of the harmful influence of intermetallic particles on mechanical properties is that they are much easier to break at tensile loads than the aluminium matrix or small silicon particles (if modified). Increased iron content may increase the hardness of the alloy since iron-based phases achieve higher hardness compared to the primary α phase [12,13].

Copper in this system lowers the liquidus temperature and worsens ductility, improves machinability, and thermal conductivity. The biggest disadvantage is the reduced resistance to corrosion (inter-crystalline type of corrosion), where the main catalyst is the presence of copper and the impact of the environment. There are no ternary compounds in the system [14,15].

The AlSi9Cu3 alloy is characterized by a high degree of mobility of hardening element (Cu) atoms. The natural ageing process proceeds slowly and spontaneously at ambient temperature and the final properties are achieved in more than 100–150 h. After this time, a partially saturated phase α (Al) is formed in the alloy and the strength and hardness of the castings are increased due to spontaneous ageing. In the ageing process, it comes to diffusion of the additive element into the microscopic regions rich in this element and to nucleation of the new phase in them. The growth of these embryos results in coherent precipitates that are referred to as GP zones. Coherence means that these areas are part of a crystalline grid of solid solution, which deforms the grid and induces internal stresses in it, thereby increasing the strength and hardness of the alloy. Artificial ageing temperatures are selected in the range 150 to 200 °C depending on the alloy, and the hardening time is from 2 to 10 h. The properties of the alloy after hardening depends on the hardening element content [12,16,17].

## 2. Materials and Methods

The secondary AlSi9Cu3 (A226) cast alloy was used to perform the experimental work. The alloy is characterized by medium mechanical properties, good strength at elevated temperatures, and good workability. The alloy has a good running quality and a low tendency to form shrinkages. A dominant part of its usage are castings for the automotive industry that are cast mainly by pressure casting: Cylinder heads and engine blocks, crankshaft cabinets, and other components. The alloy is applied also in the electro-technical industry in various components of electric motors [18].

The alloy was prepared in the form of ingots with a total batch weight of 100 kg. To investigate the effect of a higher iron content upon multiple remelting, iron in wt% increased from the original value of 1.08 wt% to approximately 1.4 wt%. Targeted “contamination” (above the value permitted by EN 1706) of the alloy took place at 750 ± 5 °C by adding AlFe10 master alloy. The intention was to create large iron-based intermetallic phases in the microstructure, which will be subjected to the effect of multiple remelting. The newly formed alloy with higher iron content was used as a reference alloy with the designation D1 for the next experimental procedure. The chemical composition of the primary AlSi9Cu3 alloy was obtained from the standard (EN 1706), the secondary aluminum alloy (experimental material before addition of Fe) and AlSi9Cu3 alloy (D1—after addition of Fe) according to the results when using arc spark spectroscopy (Q2 ION, Bunker, Kalkar, Germany) are shown in Table 1. Its chemical composition and the calculated critical iron content for D1 alloy according to Equation (1) are given in Table 1.

The alloy melting was carried out in an electric resistance furnace in a steel crucible and it consisted of pouring ingots into the prepared metal molds. A protective graphite coating was applied to prevent direct contact of the aluminium melt with the steel crucible, tools, and mold. After solidification and cooling, these ingots were used as a charge for the following melting without further chemical treatment. This process was repeated six times. After each second melt, samples were cast for selected mechanical properties (static tensile test and Brinell hardness test for 12 pieces) and metallographic evaluation, samples with the designation D3 (after the third remelting), D5 (after the fifth remelting), and D7 (after the seventh remelting) were used. Table 2 shows the changes in wt% of the selected elements and the level of critical iron (calculated according to Equation 1) in the investigated alloys D3, D5, and D7. Chemical composition was measured by arc spark spectroscopy. As a result of the melting, a significant increase of the iron content (about 12%) occurred at the alloy D7 compared to the reference alloy D1. This increase is likely to be due to the insufficient treatment of the steel crucible with paint after each melting, which could cause the contamination of melting with the elements mentioned above, since the aluminium alloy is capable of dissolving iron from unprotected steel tools. All test samples were made under the same conditions. The casting temperature was in the range of 750–760 °C and the temperature of the metal mold was set at 100 ± 5 °C. The melt was not vaccinated, modified, or refined. Before casting, only the oxide films were removed mechanically [19].

The samples (1 cm × 1 cm) for metallographic observations were prepared by standard metallographic procedures (wet ground, polished with diamond pastes, finally polished with commercial fine silica slurry (STRUERS OP-U, Prague, Czech Republic) from selected tensile specimens (after testing). The microstructure of experimental material was studied using optical microscope Neophot 32 and SEM observation with EDX analysis using scanning electron microscope VEGA LMU II (Tescan, Brno, Czech Republic) linked to the energy dispersive X-ray spectroscopy (EDX analyser Brucker Quantax, Bunker, Kalkar, Germany). Samples were etched by a standard reagent (0.5% HF). Some samples were also deep etched for 30 s in a HCl solution in order to reveal the three-dimensional morphology of the eutectic silicon and intermetallic phases. The specimen preparation procedure for deep etching consists of dissolving the aluminum matrix in a reagent that will not attack the eutectic components or intermetallic phases. The residuals of the etching products should be removed by intensive rinsing in alcohol. Each sample was subjected to measuring the length of the Al_5_FeSi ferric phase at 500× magnification.

The process of crystallization of alloys with different degrees of remelting was evaluated by thermal analysis. A K-type (NiCr-Ni) thermocouple placed in the center of a cylindrical metal mold with a diameter of 34 mm and a height of 50 mm was used during the measurement. Values were recorded in LabView 2 Hz software (version 18.5, National Instruments, Austin, TX, USA).

The tensile test was performed in accordance with the STN EN ISO 6892-1 standard on testing machine WDW 20 (Jnkason, Jinan, China) with a maximum load of 20 kN and a constant crosshead feed rate of 2 mm/min. Samples were made from the casting with turning and milling operations. The Brinell hardness test was performed according to STN EN ISO 6506-1 on testing machine INNOVATEST NEXUS 3002XLM-INV1 (Innovatest, Borgharenweg, Netherlands) with a load of 125 kp (1226 N), 5 mm diameter ball and a dwell time of 15 s. The Brinell hardness value at each state was obtained as the average of at least six measurements. Samples were taken from the front surfaces of the torn bars from the static pull test.

Each alloy was gradually evaluated in three different states, in the cast state (CS—no additional heat treatment performed). Structural analysis and mechanical testing of cast samples were performed within 24 h of casting. After natural ageing (NA—about 160 h at 20 °C) and after heat treatment (AA—T5 artificial ageing at 200 ± 5 °C for 4 h and cooling by water to 60 ± 5 °C).

## 3. Results

### Evaluation of Microstructure

#### 3.1.1. Cast State

On the reference, the microstructure of alloy D1 containing 1.416 wt% Fe consists of dendrites α-phase (light grey), eutectic Si, and intermetallic Fe-rich phases (Figure 1a). Iron-based intermetallic phases precipitate in the interdendritic and intergranular regions as platelets (appearing as needles in the metallographic microscope). Iron particles are evenly distributed in the interdendritic regions of α-phase. After the third remelting, there was a change in their reduction, possibly due to partial fragmentation of the plates and needles of intermetallic phase (Figure 1b). Due to further remelting and increasing wt% of Fe (Table 1 and Table 2), the number and dimensions of the iron phases have increased and are present exclusively in the needle morphology (Figure 1c,d). The change in eutectic Si morphology began to manifest after the fifth remelting. Smooth and large plate-like particles (Figure 1a, deep etch.) of eutectic Si (typical of the reference alloy D1 and the alloy after the third remelting D3) changed to coarse and polyedra morphology due to further remelting (Figure 1d, deep etch).

Thermal analysis of the examined alloys focused on the evaluation of the effect of multiple remelting on the process of crystallization of iron intermetallic phases and their morphology. In all investigated alloys, the critical iron level (1) was exceeded, resulting in the formation of undesirable iron-based intermetallic phases in the alloy structure preferably before the eutectic phases (Figure 2).

Significant change of course of reaction, increase in temperature, and prolongation of iron-based intermetallic phase formation are observed gradually increasing the number of remelting. From the curves of the first derivative, a gradual increase of the peak of the curve is visible in the region characterized by the formation of iron-rich phases, indicating that at higher wt% Fe is released at a greater amount of latent heat during the crystallization of these phases. The process of multiple remelting did not significantly affect the temperature changes of the other structural components. (Table 3).

#### 3.1.2. Natural Ageing

The ASM Handbook [20] claims that two types of iron-rich phases (α - Al_15_(FeMn)_3_Si_2_ and β - Al_5_FeSi) may occur in AlSi9Cu3 alloys. The structure of the reference alloy D1 and alloy D3 after the third remelting is formed by intermetallic phases β - Al_5_FeSi, whose morphology, due to natural ageing has not changed (Figure 3a). The SEM image shows the regular distribution of thin β - Al_5_FeSi needles with a monoclinic crystal structure, which are interlaced with silicon plates. The EDX of D1 reference alloy after natural ageing particles is shown in Figure 4. Al, Si, Fe, and only a small amount of Mn was identified in the needle particle, confirming the presence of the β - Al_5_FeSi phase. Table 4 shows the mean measured length of β - Al_5_FeSi phase needles for each alloy after natural ageing. The particles of eutectic Si are present in the unmodified form of large hexagonal plates with visible twinning, which are in the form of needles (Figure 3b). On the surface of the eutectic Si plates there are protruding irregular steps related to the twinning of the Si crystal.

The structures of alloys after the fifth remelting D5 and seventh remelting D7 after natural ageing are characterized (as cast state) by the presence of β - Al_5_FeSi phases. The β-Al_5_FeSi phases are present (by the metallographic microscope observation) in thicker needle morphology (locally occurring also in the form of platelets) with an increase in their lengths (Table 4), which are even distributed in the aluminum matrix (Figure 5a). Due to the higher number of remelts, eutectic Si crystallized in the form of thick hexagonal plates with undirected distribution or polyedra grain form (Figure 5b). Irregular multiple twinning can be seen on hexagonal plates.

The experimental material belongs to Al-Si-Cu materials. Cu is in Al-Si-Cu cast alloys present primarily as phases: Al_2_Cu, Al-Al_2_Cu-Si, as well as in the presence of Fe type Al_7_FeCu [21]. Intermetallic phases that contain a rich amount of copper and are included in all experimental alloys are primarily eliminated in the vicinity of eutectic silicon grains and, in particular, iron phase needles (Figure 6). The hardening phases of Al_2_Cu can be eliminated in two forms, namely in the so-called “blocks” wherein the intermetallic compound contains about 40 wt% Cu. The second form is a finely released eutectic containing about 24 wt% copper released with aluminum. Based on the EDX point analysis (Figure 5a), it can be concluded that the phase Al_2_Cu observed is released in the form of a fine eutectic.

#### 3.1.3. Artificial ageing—T5

The effect of artificial ageing led to a local refinement and shortening of the iron phase needles compared to natural ageing (Table 5). As a result of artificial ageing, the shape of the eutectic silicon on the alloy has changed on reference alloy D1, after the third D3 and the fifth D5 remelting (Figure 7a,b,c). From the hexagonal plate morphology (typical of alloys after natural ageing), the edges grew and gradually arched and thinned, between the dimensional shape of β - Al_5_FeSi phases. In the alloys D1 and D3 (Figure 7a,b), branching and splitting of eutectic Si plates can also be seen. In the alloy after the seventh remelting of D7, artificial ageing caused only local growth of the edges with arching without subsequent thinning (Figure 7d).

We observed intermetallic phases Al_7_FeCu_2_ (L + Al_7_FeCu_2_ → (Al) + Al_2_Cu + Al_5_FeSi) in the alloys after the fifth and, especially, the seventh remelting (Figure 8a) [22]. The D7 alloy sample, after its seventh remelting, was subjected to a linear EDX method where the composition of the individual phases present in the structure was identified (Figure 8b). The chemical composition of the excluded intermetallic phase of the needled morphology exhibits Al, Cu, and Fe elements, i.e., it can be the Al_7_FeCu_2_ phase. The second observed phase (on the right side) is mostly composed of Al and Cu elements, confirming the presence of the Al_2_Cu (containing about 24 wt%—fine eutectic) hardening phase.

#### 3.1.4. Mechanical Properties

The results of the tensile and elongation tests are graphically shown in Figure 9. The numeric values represent the arithmetic mean of four measurements for each state. The maximum tensile strength R_m_ = 193 MPa (Figure 9a) and the elongation A_50_ = 1.3% (Figure 9b) of cast alloys were achieved after the third remelting D3. Due to the subsequent remelting there was a rapid decrease of the two investigated characteristics to the minimum values achieved on the alloy after the seventh remelting D7 (R_m_ = 144 MPa, A_50_ = 0.5%). Degradation of tensile strength and elongation can be attributed to an increasing iron content, and hence increased iron phases (β - Al_5_FeSi) in the offset needle morphology in the structure with a higher number of remelting, thereby disturbing the structure. The tensile strength on the reference alloy D1 increased after natural ageing R_m_ = 182 MPa and after artificial ageing R_m_ = 186 MPa (an increase of 4% compared to cast state). On the alloy after the third remelting D3, due to natural and artificial ageing an increase in both cases was recorded by about 1% compared to the cast state. A fundamental change occurred on the alloys after the fifth D5 and after the seventh remelting D7. Artificial ageing resulted in an increase in tensile strength to Rm value = 174 MPa (an increase by 18% compared to the cast state) on the alloy after the fifth remelting D5 and on the alloy after seventh remelting D7 to R_m_ value = 165 MPa (increase by 15% compared to the cast state), while the results of tensile strength of the alloys after the fifth remelting D5 and after the seventh remelting D7 after natural ageing have not improved and the resulting tensile strength values ranged at the same level as in the cast state. After applying the natural and artificial ageing, the decrease of elongation was minimized in all cases. The red line in the graphs indicates the minimum values required by the standard (EN 17 06) for the gravity cast AlSi9Cu3 alloy. The minimum required elongation according to the EN 1706 standard (A_50_ = 1%) was achieved only on the reference alloy and after the third remelting in the cast state and after natural ageing.

On the basis of the hardness results (Figure 10) it can be stated that by the effect of remelting, these did not change significantly. Artificial ageing has led to a single significant change in hardness only for alloys with a higher number of remelting. The increase in hardness for alloys after the fifth D5 and after the seventh remelting D7 may be due to the presence of a large amount of iron phases and possibly also sludge phases in the structure of the alloys characterized by increased hardness. The hardness results on all samples comply with the EN 1706 standard.

## 4. Discussion

As a result of remelting the examined AlSi9Cu3 alloy, it was assumed that there were negative changes in tensile strength and elongation after the third remelting. This phenomenon is a consequence of changes in the morphology of the excluded β - Al_5_FeSi iron phase. The presence of undesirable needle morphology in alloys with a higher number of remelting results in the distortion of the structure by formation of pre-cracks and it is assumed, for example, to increase the porosity. Along with the increasing wt%, iron (Table 2) also increases the number and dimensions of β - Al_5_FeSi phase. A larger number of these phases degrade in the strength and elongation, as they are characterized by higher brittleness and at the tensile load they are more susceptible to fracture than the aluminum matrix [8,9,16]. The significant occurrence of oxidic films has not been observed with an increasing amount of recycled material (see microstructure). On the basis of indirect methods such as (density index) a slight increase in gas amount was measured for alloys with a higher number of remelting and it is possible to assume fixation of oxide films on the formed bubbles, which may also negatively affect the mechanical properties [23,24,25].

The application of artificial ageing has led to improved mechanical properties in all alloys. On the alloy after the fifth remelting D5, the improvement is by about 18%, and on the alloy after seventh remelting D7 it is by about 15%. Participating in the above increase has been by the occurrence of the shortening of the dimensions of β - Al_5_FeSi phase needles, compared to the samples aged naturally and, probably, a greater incidence of coherent and semi-coherent phases formed due to artificial ageing. The formation and presence of these phases causes deformation of the basic matrix and thus affects the mechanical and physical properties of the alloy [15,16,17,26]. Small precipitates incipient by age-hardening were invisible in the optical microscope and electron microscope so it is necessary to observe them using TEM microscopy [17].

Improving tensile strength through natural ageing is only observed in alloys with a lower number of remelting (D1 and D3), the efficiency of which is comparable to artificial ageing. A fundamental change occurred in alloys after the fifth and seventh remelting, when the tensile strength results reached the level of the cast state.

## 5. Conclusions

Investigating changes in mechanical properties and structural changes due to AlSi9Cu3 alloy remelting can be termed as critical after the fourth remelt. After the fourth remelting, significant degradation of the structural components is observed. The iron-rich phases of alloys with five and more remelting crystallize in the form of thicker needles with significantly larger average lengths. Due to the increasing number of remelts, a significant destruction (degradation) of eutectic silicon from predominantly regular hexagonal plates to irregular polyhedral-type morphology occurred. The change of the structural components had a fundamental effect on the decrease of mechanical properties and especially the ductility of experimental alloys. The application of natural ageing had no major effect on the individual structural components, and similar mechanical properties were achieved compared to the casted state. The application of artificial ageing resulted in a reduction of the iron-rich phase average needle lengths and partial spheroidization of eutectic silicon, resulting in an increase of mechanical properties in all test alloys.

The use of alloys with a higher number of remelting, let us say with higher contamination for the production of dimensionally challenging molded castings is possible by using a suitable method of elimination. As a suitable way to achieve the desired characteristics, the application of T5 heat treatment was demonstrated by artificially ageing.

## Figures and Tables

**Figure 1 materials-13-00575-f001:**
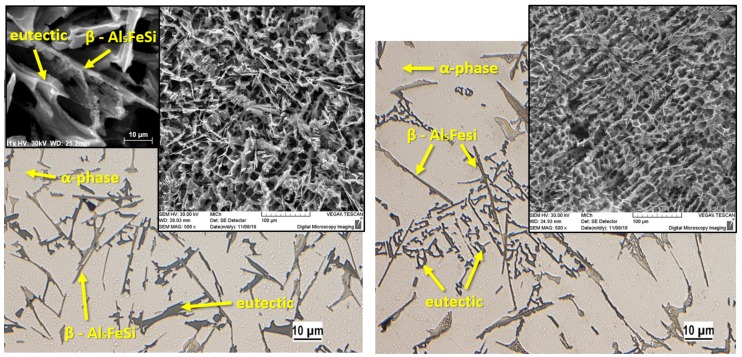
Microstructure of experimental AlSi9Cu alloy depending on the number of remelting in cast state; optical microscopy, etch 0.5%; deep etch, SEM. (**a**) D1 alloy; (**b**) D3 alloy; (**c**) D5 alloy; (**d**) D7 alloy.

**Figure 2 materials-13-00575-f002:**
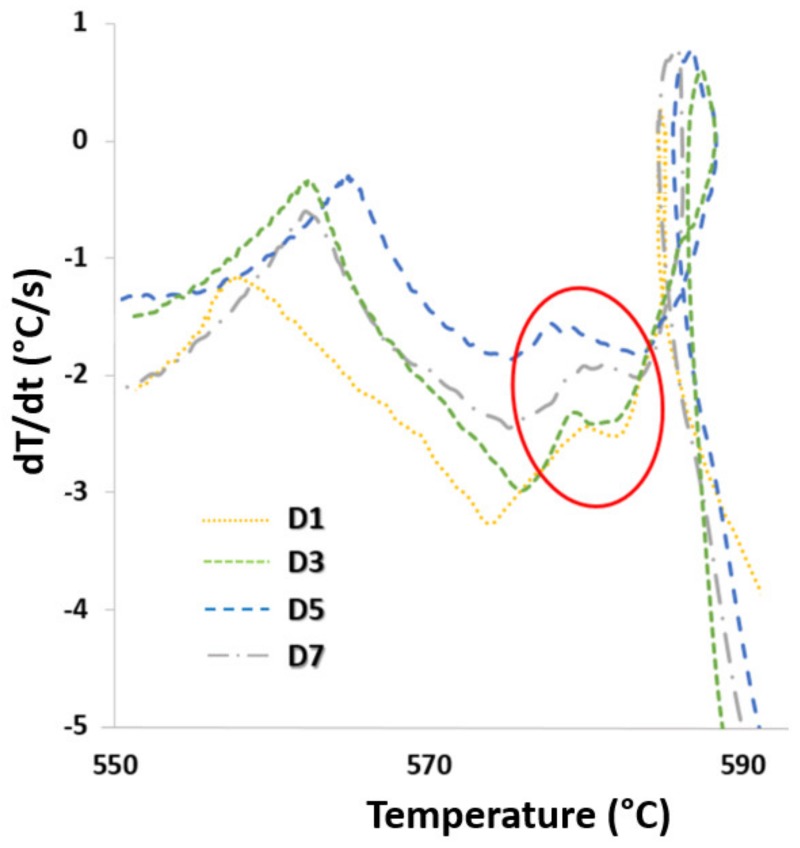
Iron reach phases detected using first derivative curves of AlSi9Cu3 alloys in various degrees of remelting.

**Figure 3 materials-13-00575-f003:**
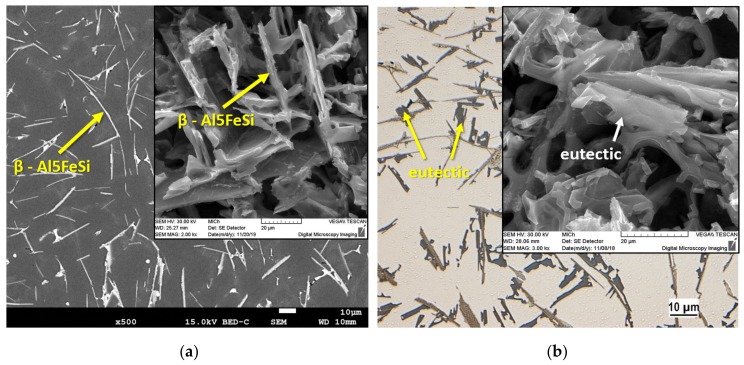
Microstructure of D1 reference alloy after natural ageing, SEM. (**a**) Deep etch, SEM; (**b**) optical microscope, deep etch, SEM.

**Figure 4 materials-13-00575-f004:**
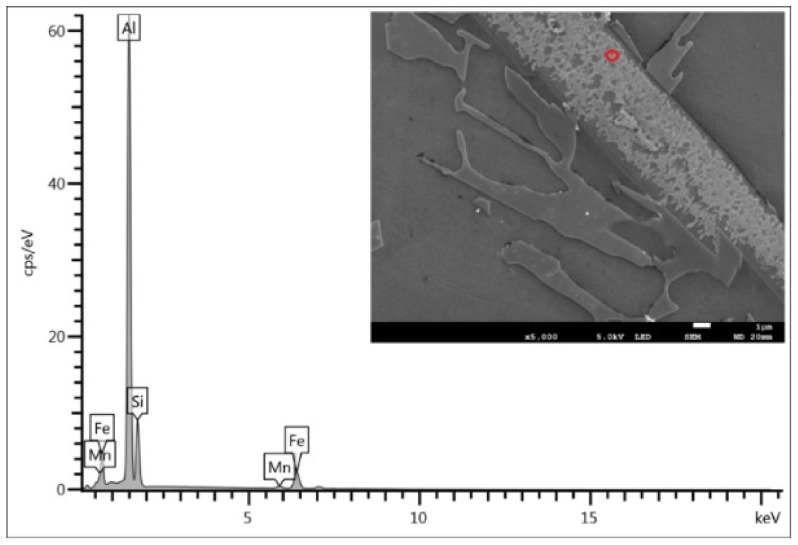
EDX point analysis of β - Al_5_FeSi particles of D1 reference alloy after natural ageing, SEM.

**Figure 5 materials-13-00575-f005:**
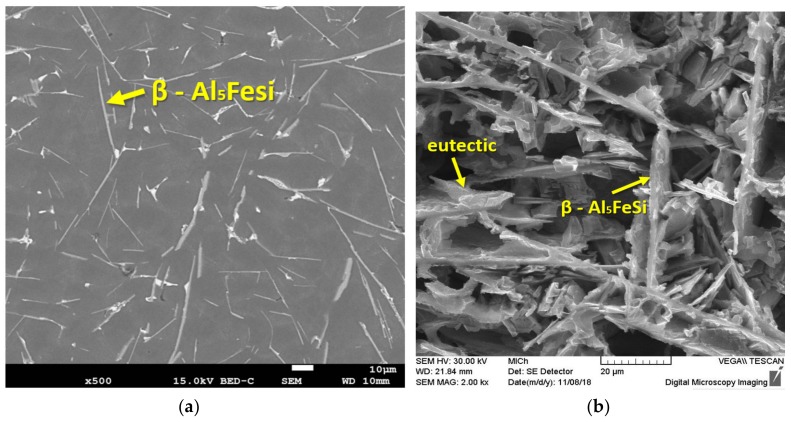
Microstructure of D5 alloy after natural ageing. (**a**) SEM; (**b**) deep etch SEM.

**Figure 6 materials-13-00575-f006:**
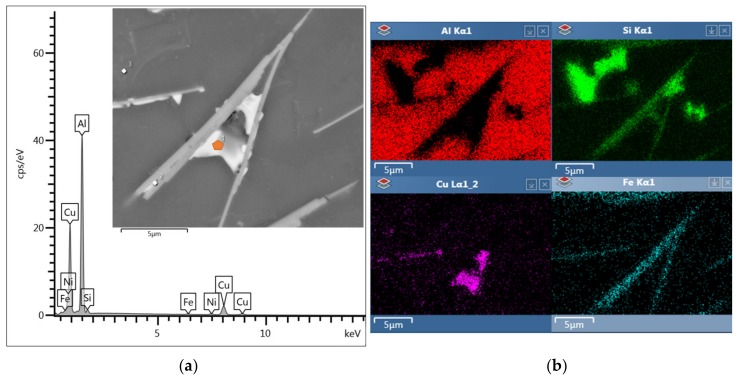
Copper and iron rich area of precipitation, alloy after 5th remelting D5 after natural ageing. (**a**) EDX point analysis of Al_2_Cu particles, SEM; (**b**) mapping elements of various structural components.

**Figure 7 materials-13-00575-f007:**
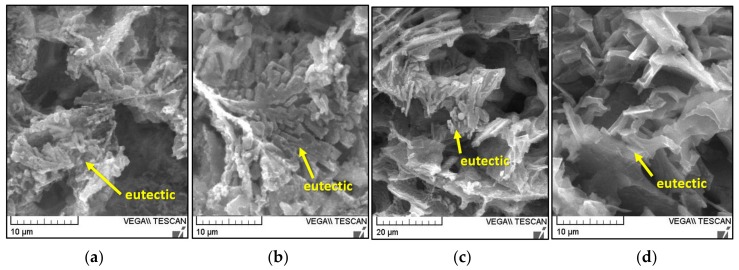
Eutectic silicon after deep etching of experimental alloys after artificial ageing, SEM. (**a**) D1 alloy; (**b**) D3 alloy; (**c**) D5 alloy; (**d**) D7 alloy.

**Figure 8 materials-13-00575-f008:**
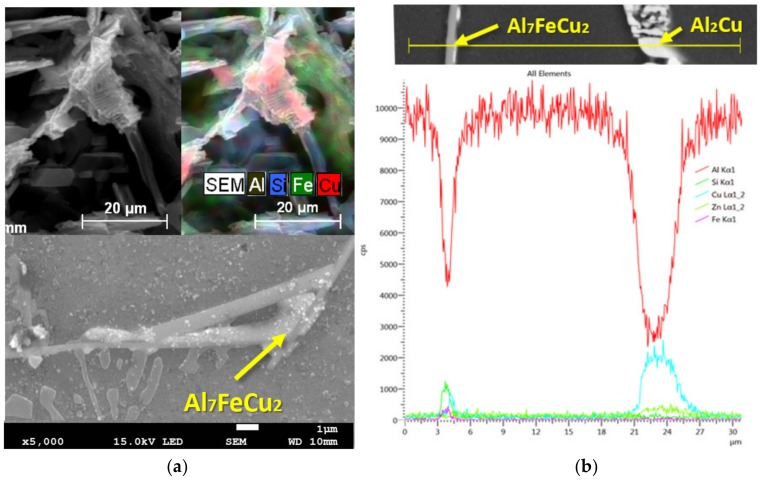
Microstructure of D7 alloy after artificial ageing. (**a**) Al_7_FeCu_2_ phase, deep etch, SEM; (**b**) line EDX analysis of D7 alloy after artificial ageing, SEM.

**Figure 9 materials-13-00575-f009:**
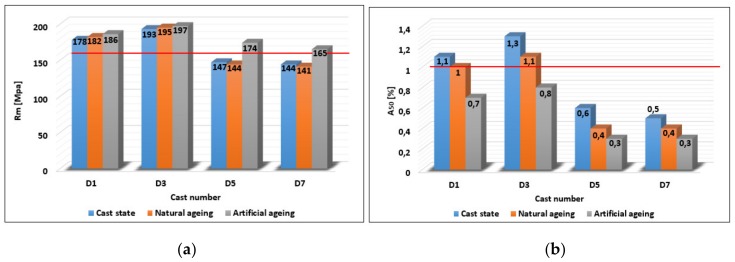
Relationship between (**a**) tensile strength and (**b**) elongation and melt number and alloy state.

**Figure 10 materials-13-00575-f010:**
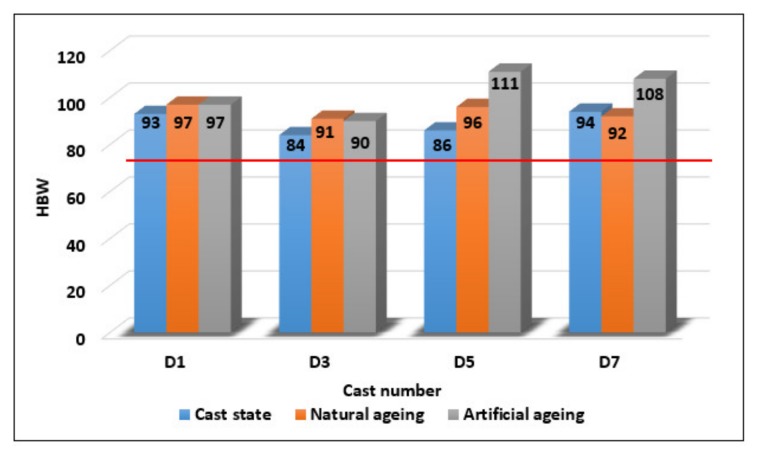
Relationship between Brinell hardness and melt number and alloy state.

**Table 1 materials-13-00575-t001:** Chemical composition of primary, secondary, and D1 AlSi9Cu3 alloy (wt%).

Elements	Si	Fe	Cu	Mn	Mg	Ni	Zn	Ti	Cr	Fe_crit_
Primary AlSi9Cu3 (EN 1706)	8.0–11.0	0.6–1.1	2.0–4.0	0.55	0.15–0.55	0.55	1.20	0.20	0.15	–
Secondary AlSi9Cu3	9.563	1.081	2.206	0.184	0.426	0.092	1.160	0.038	0.027	0.667
D1 alloy	9.441	1.414	2.174	0.174	0.429	0.090	1.158	0.035	0.024	0.658

**Table 2 materials-13-00575-t002:** Chemical composition of D3, D5, and D7 alloys (wt%).

Elements	Si	Fe	Cu	Mn	Mg	Ni	Zn	Ti	Cr	Fe_crit_
D3 alloy	9.316	1.475	2.114	0.186	0.423	0.097	1.157	0.037	0.043	0.649
D5 alloy	9.313	1.51	2.094	0.181	0.407	0.115	1.144	0.033	0.061	0.648
D7 alloy	9.286	1.612	2.097	0.187	0.394	0.133	1.173	0.031	0.103	0.646

**Table 3 materials-13-00575-t003:** Characteristic temperatures of investigated alloys.

Alloy	T_liq_ (°C)	T_Al5FeSi_ (°C)	T_Al-Si_ (°C)	T_Al-Cu_ (°C)	T_sol_ (°C)
D1	632	578	569	515	478
D3	629	582	571	517	476
D5	628	584	574	520	477
D7	630	586	573	518	474

**Table 4 materials-13-00575-t004:** Measurement results of length of iron-based particles after natural ageing.

Alloy	D1	D3	D5	D7
Average length of iron phases (μm)	55.7	27.7	71.4	89.4

**Table 5 materials-13-00575-t005:** Measurement results of length of iron-based particles after artificial ageing.

**Alloy**	**D1**	**D3**	**D5**	**D7**
Average length of iron phases (μm)	37.8	26.3	55.1	72.6

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
