# Peer review of "The Influence of Remelting on the Properties of AlSi9Cu3 Alloy with Higher Iron Content"

_materials, 2020, doi:10.3390/ma13030575_

Round 1

Reviewer 1 Report

The article presents the influence of remelting on the properties of AlSi9Cu3 alloy with higher iron content.

Dealing with recycling of Al alloys, the topic is pretty interesting and timely. 

However, authors should take the following suggestions into consideration: 

1)The use of the English language needs extensive revision. Some sentences actually don't make sense, while many typographical errors have not been noticed by the authors.

2) The introduction section is quite up to date, however, a clear paragraph about the actual need of this specific effort to be done, could be added.

3)  The analysis of the SEM images is pretty basic and it is based on pseculations. A further support from the literature could be added. 

4) The tensile strength and hardness should have a standard deviation value. 

5) The discussion section looks more like a technical report, rather than a scientific paper. Extra analysis is of vital need.

6) The number of references (20) could be increased. 

Author Response

Dear Reviewer, 

All updated and modified information (at the reviewer's request) in the text are marked by yellow.

Figures 1 and 3b have been redesigned at the reviewer's request.

(Contrast and brightness have been adjusted to differentiate between phases. And in Figure 1, 50 µm images were replaced with higher magnification (10 µm) images). 

The paper was translated, checked and edited by SARJA company (translator Mr. Gažo) with the address Framborská 58, 010 01 Žilina, Slovakia before the first submission. Our department has been cooperating with mentioned company for several years and there has never been a bigger problem with translation From which we conclude that the article should be translated into English with the required quality. Correcting English by another translation company is problematic with respect to the current date and Christmas holidays.

1.English has been additionally reviewed.

2. A paragraph describing the real need for the issue has been added to the introduction.

3. SEM image analysis has been added.

4.Tensile strength - in the D3 alloy (after the third remelting) there was an unexpected slight increase in the values associated with the structure, partial fragmentation of the ferric phase needles was observed, resulting in a shortening of the average lengths. The shortening of the average needle lengths in this case resulted in a slight increase in tensile strength. Subsequent increase in the number of remelts, the ferric phases were segregated in thicker needles and there was no fragmentation, increasing their average length, and this led to an expected decrease in mechanical properties.

Hardness - Shorter average Fe phase lengths for the D3 alloy resulted in a slight decrease in hardness, the iron rich phases have a higher strength compared to the eutectic and the aluminum matrix. For alloy D5, there was a slight increase of hardness when needle length increased, but due to Si degradation, the hardness value was lower compared to alloy D1. In the D7 alloy, although the greatest Si degradation occurred as a result of the remelting process, the Fe needles reached the greatest average lengths, thereby increasing the hardness value for the alloy. The application of natural and artificial ageing to D1 and D3 alloys resulted in approximately the same increase in hardness. In the D5 alloy, the application of natural ageing led to an increase but was no longer as effective as artificial ageing. In the D7 alloy, natural ageing proved to be ineffective and only increase of hardness occured in the case of artificial ageing.

5. The conclusion has been reformulated.

6. References have been added

Thank you for your consideration!

Reviewer 2 Report

The article mainly explores the effects of remelting times on microstructure and mechanical properties of AlSi9Cu3 alloy with higher iron content. Microstructure observation, hardness measurement, and tensile tests were conducted in the study. Most of the content in the paper are understandable after reading, but it visibly exhibits some doubts as below.

In the abstract, some content is repeated. Is there any oxide in the structure after remelting? Why are the trends inconsistent in hardness and strength as shown in Figures 9 and 10? Line 229, description of the content appears to be incorrect. Overall, in my opinion, the article in the present form is not suitable for publication in journal of ‘materials’.

Author Response

Dear Reviewer,

All updated and modified information (at the reviewer's request) in the text are marked by yellow.

Figures 1 and 3b have been redesigned at the reviewer's request.

(Contrast and brightness have been adjusted to differentiate between phases. And in Figure 1, 50 µm images were replaced with higher magnification (10 µm) images).

The paper was translated, checked and edited by SARJA company (translator Mr. Gažo) with the address Framborská 58, 010 01 Žilina, Slovakia before the first submission. Our department has been cooperating with mentioned company for several years and there has never been a bigger problem with translation From which we conclude that the article should be translated into English with the required quality. Correcting English by another translation company is problematic with respect to the current date and Christmas holidays.

1. The repetitive part was removed from the abstract and information about the techniques used in the structural analysis was added (so that the abstract contains approximately 200 words).

2. The evaluation of oxides has not been closely investigated in the present article, but a significant occurrence of oxdic films has not been observed with an increasing amount of recycled material (see microstructure). Additional evaluation was carried out in a large-scale experiment. On the basis of indirect methods such as (Density index) a slight increase in gas amount was measured for alloys with a higher number of remelting and it is possible to assume fixation of oxide films on the formed bubbles.

3.

Tensile strength - in the D3 alloy (after the third remelting) there was an unexpected slight increase in the values associated with the structure, partial fragmentation of the ferric phase needles was observed, resulting in a shortening of the average lengths. The shortening of the average needle lengths in this case resulted in a slight increase in tensile strength. Subsequent increase in the number of remelts, the ferric phases were segregated in thicker needles and there was no fragmentation, increasing their average length, and this led to an expected decrease in mechanical properties.

Hardness - Shorter average Fe phase lengths for the D3 alloy resulted in a slight decrease in hardness, the iron rich phases have a higher strength compared to the eutectic and the aluminum matrix. For alloy D5, there was a slight increase of hardness when needle length increased, but due to Si degradation, the hardness value was lower compared to alloy D1. In the D7 alloy, although the greatest Si degradation occurred as a result of the remelting process, the Fe needles reached the greatest average lengths, thereby increasing the hardness value for the alloy. The application of natural and artificial aging to D1 and D3 alloys resulted in approximately the same increase in hardness. In the D5 alloy, the application of natural aging led to an increase but was no longer as effective as artificial aging. In the D7 alloy, natural aging proved to be ineffective and only increase of hardness occured in the case of artificial aging.

4.Incorrect image numbering has been fixed.

Thank you for your consideration!

Reviewer 3 Report

The manuscript is well written.

Line 25/26: The keywords should be more paper related and not as general as microscopy, SEM etc.

Line 92: usedas – used as

Line 153: Figure 1, a, b, c, d resolution 50µm, its almost impossible to see any difference within the phases, beside the color. Some of the arrows seem to point at the same phases. Therefore, it would be good to improve the visibility of the different phases and the areas where the arrows point. A presentation in that way, doesn’t add any value to the manuscript.

Line 182: Figure 3, a, b, it is the same as for Figure 1, the figures with a resolution of 10µm don’t add any value to the manuscript.

Author Response

Dear Reviewer,

All updated and modified information (at the reviewer's request) in the text are marked by yellow.

The paper was translated, checked and edited by SARJA company (translator Mr. Gažo) with the address Framborská 58, 010 01 Žilina, Slovakia before the first submission. Our department has been cooperating with mentioned company for several years and there has never been a bigger problem with translation From which we conclude that the article should be translated into English with the required quality. Correcting English by another translation company is problematic with respect to the current date and Christmas holidays.

1. Keywords have been amended.

2. usedas – used as - Corrected

3. The optical microscope images in Figures 1 and 3 were changed. Contrast and brightness have been adjusted to differentiate between phases. And in Figure 1, 50 µm images were replaced with higher magnification (10 µm) images. I hope the changes will be sufficient.

Thank you for your consideration!

Round 2

Reviewer 1 Report

The authors added some new data in the revised manuscript, however, the work still remains scientifically incomplete.

The insufficient use of the English language also appears as an obstacle to my decision to accept the publication of this work in "Materials".

Reviewer 2 Report

The authors have responded appropriately and done some work to improve the paper. Thereby, I can agree with the revisions to my prior comments.